# Protective Effects of Polyphenols against Ischemia/Reperfusion Injury

**DOI:** 10.3390/molecules25153469

**Published:** 2020-07-30

**Authors:** Martina Cebova, Olga Pechanova

**Affiliations:** Centre of Experimental Medicine Slovak Academy of Sciences, Institute of Normal and Pathological Physiology, 81371 Bratislava, Slovakia; olga.pechanova@savba.sk

**Keywords:** myocardial infarction, polyphenols, cardiovascular diseases, nitric oxide, ROS, phytochemicals, ischemia/reperfusion injury

## Abstract

Myocardial infarction (MI) is a leading cause of morbidity and mortality across the world. It manifests as an imbalance between blood demand and blood delivery in the myocardium, which leads to cardiac ischemia and myocardial necrosis. While it is not easy to identify the first pathogenic cause of MI, the consequences are characterized by ischemia, chronic inflammation, and tissue degeneration. A poor MI prognosis is associated with extensive cardiac remodeling. A loss of viable cardiomyocytes is replaced with fibrosis, which reduces heart contractility and heart function. Recent advances have given rise to the concept of natural polyphenols. These bioactive compounds have been studied for their pharmacological properties and have proven successful in the treatment of cardiovascular diseases. Studies have focused on their various bioactivities, such as their antioxidant and anti-inflammatory effects and free radical scavenging. In this review, we summarized the effects and benefits of polyphenols on the cardiovascular injury, particularly on the treatment of myocardial infarction in animal and human studies.

## 1. Introduction

Despite current optimal treatments, cardiovascular diseases (CVDs) are the most common global causes of morbidity and mortality. Of total cardiovascular deaths, coronary artery disease is the leading cause at 60%, of that acute myocardial infarction (MI) comprises 25% [1]. Increasingly, these diseases occur in people of working age, requiring long-term treatment and/or limiting the ability to work due to disability, with serious economic and social consequences. A poor diet, smoking, alcohol, lack of exercise, obesity, and everyday stress brings with it a number of risk factors that can manifest as various diseases. Although advances in diagnostic and therapeutic drug developments have largely benefited patients with cardiovascular diseases, the serious side effects of some medications may limit their applications [2,3].

Over the past 20 years, scientists have been interested in polyphenols. The main reasons for this interest are the antioxidant properties of polyphenols and their probable role in the prevention of various diseases, such as cardiovascular and neurodegenerative diseases, cancer, and diabetes mellitus, associated with oxidative stress [4,5,6]. The popularity of natural compound treatments for CVDs has increased not only in Asia but also in some Western countries, including Europe, the USA, and Australia [7,8]. Polyphenols are biomolecules that originate from plants and have antioxidant and anti-inflammatory effects [9], which can modulate the activity of a wide range of enzymes and cell receptors [10]. In this way, in addition to having antioxidant properties, polyphenols offer several other specific biological actions that remain poorly understood. In this review, we mainly focused on recent experimental studies exploring the effects of some polyphenols in a model of myocardial infarction and provided information about the proposed mechanisms involved in the cardiac effects of natural compounds.

## 2. Pathogenesis of Ischemia/Reperfusion Injury

Myocardial infarction is a serious disease caused by a sudden disruption of the oxygen supply to the heart muscle and the subsequent death of the muscle fibers. In a clinical context, MI usually occurs due to the formation of atherosclerotic plaque. When the plaque endures a rupture, the disruption of blood circulation by reduced perfusion and/or occlusion of the coronary artery causes by the formation of a thrombus. Coronary artery blockage results in myocardial infarction. Acute myocardial infarction is a potentially life-threatening condition. When the heart survives the ischemia, either adaptive mechanisms (up to 72 h, after MI) or cardiac remodeling (after more than 72 h) occurs [11,12]. Functionally, the loss of cardiomyocytes causes changes in the ejection volume and final diastolic volume, ultimately resulting in ventricular hypertrophy. When cardiac hypertrophy cannot compensate for the increased ventricular volume, it experiences ventricular enlargement and dysfunction [13].

The lack of oxygen and nutrient supply results in a series of abrupt biochemical and metabolic changes within the myocardium, which discontinues oxidative phosphorylation, leading to mitochondrial membrane depolarization, ATP depletion, and the inhibition of myocardial contractile functions. This process is worsening by the breakdown of any ATP available, resulting in ATP hydrolysis and an increase in mitochondrial inorganic phosphate. In the absence of oxygen, cellular metabolism switches to anaerobic glycolysis with reduced intracellular pH. The intracellular concentration of sodium ions is significantly increased, and this process activates the Na^+^-H^+^ ion exchanger [14]. The lack of ATP during ischemia discontinues the function of 3Na^+^-2K^+^ ATPase as a consequence of an intracellular Na^+^ overload. The reverse activation of the 2Na^+^-Ca^2+^ ion exchanger results in an intracellular Ca^2+^ overload [15] and promotes cardiomyocyte apoptosis [16,17]. Apoptosis plays a key role in cardiomyocytes’ death after myocardial infarction. The mitogen-activated protein kinase (MAPK) signaling cascade, including extracellular signal-regulated kinase (ERK), c-JunNH2-terminal kinase (JNK), and p38 mitogen-activated protein kinase (p-38 MAPK), has been identified as a main signal transduction pathway in cardiomyocyte apoptosis [18]. Increases in intracellular calcium (Ca^2+^) can induce the phosphorylation of JNK and p38-MAPK [19,20]. Moreover, the upregulation of phosho-p38MAPK increases cardiomyocyte apoptosis, while its inhibition prevents cardiomyocyte apoptosis [21,22]. Caspase-3 is a member of the caspase family that plays a role in the execution phase of the apoptotic cascade [23] and can be activated by p38 MAPK and JNK [24,25].

Following MI, cardiac cells die in response to ischemia [26]. Cardiomyocyte death begins in the subendocardium and extends transmurally over time toward the epicardium [13]. The adult mammalian heart has a limited capacity to regenerate after injury, and the lost cells are replaced by a fibrotic scar. This is followed by cardiac remodeling accompanied by changes in the geometry and morphology of the infarcted, endangered, and non-infarcted part of the myocardium, which eventually leads to impaired cardiac function. Thickening (hypertrophy) and stiffening (fibrosis) of the left ventricular wall are part of this complex process [27]. First, ultrastructural changes, including diminished cellular glycogen, relaxed myofibrils, sarcolemmal disruption, and mitochondrial abnormalities, are seen 10–15 min after the onset of ischemia [28,29]. However, the identification of myocyte necrosis by postmortem examination in humans takes hours, in contrast to animal models, in which biochemical evidence of myocardial cell death due to apoptosis can be detected within 10 min of induced myocardial ischemia [30]. After 20–30 min of ischemia, subendocardial cardiomyocytes in the at-risk area exhibit irreversible changes. Prolonged ischemia triggers a “wavefront” of cardiomyocyte death. Because the adult mammalian heart has a negligible regenerative capacity, repairs to the infarcted myocardium are dependent on the formation of a collagen-based scar and require the activation of a superbly orchestrated inflammatory/reparative response [28]. Inflammatory chemokines and cytokines are critically involved in the reparative process, mediating the activation of macrophages that clear the wound of dead cells and matrix debris, thereby triggering an expansion of fibroblasts and secreting a collagen-based matrix network. Matrix metalloproteinases (MMPs) stimulate the disintegration of collagen in the necrotic area, resulting in the loss of support tissue. Although scar formation protects the heart from catastrophic early complications (such as cardiac rupture), infarct healing is closely intertwined with adverse remodeling [31]. The clearance of dead cells and matrix debris activates the anti-inflammatory pathways, leading to the suppression of cytokine and chemokine signaling. The activation of the renin–angiotensin–aldosterone system and the release of transforming growth factor-β induce the conversion of fibroblasts into myofibroblasts, thus promoting the deposition of the extracellular matrix protein [22].

In addition, myocardial infarction is associated with an antioxidant deficit, which includes decreased superoxide dismutase (SOD), glutathione peroxidase (GPx), and catalase (CAT) activities, as well as increased lipid peroxidation. Indeed, reactive oxygen species contribute to worsening the prognosis after MI [32]. Treatments with antioxidants that have the potential to inhibit oxidative stress after MI have been proposed over the years [33].

During myocardial infarction, the coronary flow is interrupted, triggering early cell death in the injured zone of the myocardium. Restoring the blood supply is mandatory to salvage the ischemic myocardium and thereby reduce the damage caused by myocardial infarction. When blood with oxygen is returned to the damaged tissue, the tissue may be saved over the long term. However, in the short term, the tissue is damaged further. This phenomenon is known as ischemia/reperfusion (I/R) injury [34]. The reperfusion phase is very dynamic, with cell death present for up to 3 days after the onset of reperfusion [35]. The injury response after reperfusion is directly correlated with the duration of ischemia. Experimental studies have shown that irreversible cardiomyocyte damage occurs after 20 min of ischemia [12,36], and I/R injury can further increase the infarct size by up to 50% of the initial size [37]. Much attention has been dedicated to characterizing the molecular and cellular basis of I/R injury [38,39]. I/R injury in coronary circulation presents as microvascular dysfunction through increased capillary permeability and edema, impaired endothelial and vascular smooth muscle cells, the release of vasoconstrictor substances from the atherosclerotic lesion, and ultimately, capillary destruction and hemorrhaging [40]. The exact molecular mechanisms underlying myocardial I/R injury are complicated and have yet to be fully understood.

Reperfusion is a multifactorial complex that involves (i) generation reactive oxygen species (ROS), (ii) calcium overload, and (iii) pronounced inflammatory responses [14,41]. The primary importance of ROS production in the pathophysiology of I/R injury was first identified 40 years ago. ROSs are an integral part of many intracellular signaling pathways, and in low concentrations, they are produced in a controlled manner [42]. During reperfusion, ROSs are generated via reactivation of the electron transport chain and the accumulation of xanthine oxidase (mainly in endothelial cells) and NADPH oxidase in neutrophils [43]. Oxidative stress induces sarcoplasmic reticulum dysfunction, which causes intracellular Ca^2+^ overload and damages the cell membrane by lipid peroxidation, thereby inducing enzyme denaturation and DNA damage [44]. Several hours after the onset of myocardial reperfusion, neutrophils accumulate in the infarcted myocardial tissue in response to the release of the chemoattractant ROS, cytokines, and activated complements [45]. Programmed cell death (apoptosis) induced by ROS has also been implicated in I/R injury, although the mechanisms remain unclear. More recently, nitric oxide (NO) has been suggested as a mediator of I/R injury. However, multiple studies have shown discrepancies between the levels of nitric oxide synthases (NOSs). Most authors have observed increased inducible NOS (iNOS) levels after activating proinflammatory cytokines [12,46,47]. On the other hand, endothelial NOS (eNOS) and neuronal NOS (nNOS) activity may be increased, decreased, or remain the same [47,48,49]. NO is a molecule with various physiological roles in the cardiovascular system. The beneficial cardiovascular effects of NO depend on the modulation of coronary vascular tone, cardiomyocyte contractility, monocyte adhesion, platelet and fibroblast aggregation, and smooth muscle cell proliferation [50]. However, the overproduction of NO by iNOS activation is implicated in the deterioration of cardiac function through several mechanisms, including the overproduction of reactive oxygen species, endothelial dysfunction, and the release of inflammatory mediators. High levels of NO can interact with the superoxide to produce peroxynitrite, resulting in mitochondrial cytochrome c release and caspase activation and, ultimately, apoptosis. NO is endogenously produced in the myocardium by NOS, which is regulated in several ways, including compartmentalization, cofactor and substrate availability, and transcriptional and translational modulations [51]. Due to the long-term increase in oxidative stress, the cofactors required for the synthesis of NOS are reduced and oxidized, which may ultimately lead to a reduction of NO production due to uncoupling eNOS. The balance between the cytotoxic and cytostatic effects of NO may involve the tissue concentration of NO produced by the various isoforms of NO synthase and their interactions with the ROS. All isoforms are activated due to ischemia and probably depend on the extent of MI [52]. It has been recently proved that redox molecules derived from nitric oxide (referred to as reactive nitrogen species (RNS)) also contribute to I/R injury via oxidative and nitrosative reactions [53,54,55]. The cyclooxygenase (COX) enzyme pathway is involved in almost all aspects of health and disease, including myocardial infarction. COX has two isoforms: COX-1 is constitutively expressed in many cell types, whereas COX-2 is constitutively expressed only in certain regions but is readily induced in inflammation and cancer [56]. To date, some studies have shown the cross-communication between iNOS and the COX-2 enzyme, suggesting a key link between these enzymes in pathological conditions. During inflammation, the NO produced by iNOS contributes to the upregulation of COX-2 and thus increases its cytotoxic effects. Additionally, iNOS is also known to be co-inducted with COX-2. Both COX-2 and iNOS may contribute to the development and progression of the coronary arterial disease [57]. Some other studies have shown that peroxisome proliferator-activated receptor gamma (PPAR-γ) ligands also reduce the tissue injury associated with myocardial I/R via inhibition of the JNK/AP-1 (c-Jun amino terminal kinase/activator protein 1) pathway [58]. Moreover, the administration of a PPAR-γ agonist reduces the myocardial infarction size by up to 60% along with the release of creatinine kinase, as well as an improvement in cardiac function [59].

TLRs (Toll-like receptors) have been shown to participate in the process of myocardial infarction as innate immune factors. In the early stages of MI, necrotic cells initiate an intense inflammatory response by activating innate immune mechanisms. There is evidence suggesting that TLR-mediated pathways play a significant role in triggering the post-infarction inflammatory response by activating the nuclear factor-kappa B (NF-κB) system [60]. Among the 13 members of the TLR family, TLR4 is expressed in the heart and is markedly induced in mouse and rat infarcts, as well as in samples obtained from cardiomyopathic hearts. Recent animal studies using mouse myocardial I/R injury models have shown that TLR4 and NF-κB expression levels are significantly increased in both the ischemic zone and the potential danger region of the myocardium and that cardiac myocyte apoptosis is induced during the early period following I/R injury only [61]. Interestingly, during the late period, TLR4 and pro-inflammatory cytokine levels increase, even more, resulting in cardiac remodeling. Recent investigations have demonstrated that TLR4 deficient mice have a decreased infarct size and suppressed inflammation [62]; they also exhibit attenuated adverse remodeling following myocardial infarction [63]. These studies suggest that TLR4 signaling may critically affect the inflammatory response in myocardial infarction progression. Moreover, clinical studies have shown that TLR4 expression in peripheral blood mononuclear cells is extremely elevated in patients with myocardial infarction and even higher at the site of the plaque rupture. A six-month follow-up study has indicated higher levels of TLR4 in patients with MI due to cardiac events than in those without MI [64]. Consequently, a myocardial infarction could result in activation of the TLR4-NF-κB signaling pathway in the myocardium and, therefore, induce inflammation and cardiac dysfunction.

The activation of transcription factor NF-κB plays an important role in the ischemic myocardium. It promotes inflammatory and fibrotic responses as a progression of myocardial remodeling via the transactivation of cytokines, chemokines, and matrix metalloproteinases (MMPs). Moreover, immunohistochemical studies have demonstrated an elevation of the NF-κB-p50 subunit in the nucleus of the infarcted zone 4 days after the confirmation of MI with increased NF-κB protein levels in the myocardium. On the other hand, the administration of the proteasome inhibitor, PS-519, interrupts NF-κB activation and preserves myocardial function and thus reduces the MI size in a porcine model of ischemia-reperfusion injury [65]. The absence of the NF-κB subunit p50 also improves heart failure after myocardial infarction [66]. Furthermore, the blockade of NF-κB activity significantly improves mortality and heart failure rates through remodeling the left ventricle, as well as suppressing the expression of the proinflammatory cytokine and chemokine in the infarcted myocardial zone, thereby contributing to the reduction of myocardial fibrosis [67]. In another study, the inhibition of NF-κB-p50 by carvedilol with anti-apoptotic, anti-inflammatory, and antioxidant properties in the heart has resulted in the suppression of inflammation and the preservation of cardiac function [68].

In the myocardium, one of the key elements of redox homeostasis is the pleiotropic transcription factor Nrf2 (nuclear factor erythroid 2-related factor 2), which regulates the expression of antioxidant proteins and thus protects against the oxidative damage caused by injury and inflammation [69]. One possible antioxidant mechanism for eliminating ROS is cellular signaling Nrf2/Keap1/ARE (the nuclear factor erytheroid-derived-2-like 2 (Nrf2)-Kelch-like ECH-associated protein 1 (Keap1)-antioxidant response element (ARE)). Through its activation, ROS can be reduced, and the cell can be protected from damage. In the absence of oxidative stress, Nrf2 signaling activity is suppressed in the cytoplasm [70]. Nrf2 interacts with Keap1 and rapidly induces Nrf2 degradation. Keap1 also negatively modulates Nrf2′s functions by preventing its translocation to the nucleus in the absence of oxidative stress. Under oxidative stress, Nrf2/Keap1/ARE signaling is triggered by phosphorylation and thus allows the translocation of Nrf2 to the nucleus [71]. Several studies have shown that the phosphatidylinositol-3-kinase (PI3K/Akt) signaling pathway is also involved in the induction of phase II-dependent Nrf2/ARE antioxidant synthesis. Activated Akt binds to the Keap1 protein and dissociates Nrf2 from the cytoplasm [72]. Intracellular and mitochondrial Ca^2+^ overload begins during acute myocardial ischemia and is exacerbated during myocardial reperfusion due to the disruption of the plasma membrane, oxidative stress-induced damage to the sarcoplasmic reticulum, and mitochondrial re-energization. Large-scale alterations in Ca^2+^ activate cell death following I/R via the mitochondrial permeability transition pore, Ca^2+^/calmodulin-dependent protein kinases, or other mechanisms [14]. The production of ROS and the reduction of NO induce inflammatory cytokines, such as interleukin-6 (IL-6) [73], which directly correlates with the post-ischemic deterioration of myocardial mechanical performance and the expansion of necrotic tissue [74]. Inflammatory cytokines and chemokines are involved in the reparatory process, activating macrophages and triggering an expansion of the reparation of fibroblasts to secrete a collagen-based matrix. Although this scar can protect the heart from early cardiac rupture, infarct healing is related to adverse remodeling in both the infarcted and the non-infarcted zones of the myocardium.

SIRT 1 (silent mating-type information regulation 1) appears to have a prominent role in cardiovascular biology; in preclinical models, it promotes a variety of physiological effects. It has been shown, for example, to protect the heart against I/R injury, hypertrophy, and cardiomyocyte apoptosis. SIRT1 overexpression leads to reduced myocardial hypertrophy, interstitial fibrosis, and oxidative stress [75]. These changes have been associated with a significant improvement in cardiac function, as assessed by the ejection fraction and fractional shortening. Moreover, SIRT1 ameliorates endothelial function via the activation of eNOS and prevents macrophage foam cell formation and the calcification of vascular smooth muscle cells. Subsequently, studies on genetically engineered mouse models have demonstrated that Sirt1 exerts atheroprotective effects by activating eNOS or diminishing the NF-κB activity in endothelial cells and macrophages. Another report has shown the protective role of Sirt1 in vascular smooth muscle cells against DNA damage, medial degeneration, and atherosclerosis [76].

Post-infarction remodeling is dependent not only on the size of the acute MI but also on the quality of the repair responses. Dead cardiomyocytes and a damaged extracellular matrix coordinate activation of the cytokine cascade and thus initiate the inflammatory response in the myocardium.

## 3. General Characteristic of Polyphenols

Polyphenols represent a large group of bioactive compounds found in foods, such as fruits, vegetables, cereals, herbs, spices, legumes, nuts, olives, chocolate, tea, coffee, and wine, which exhibit antioxidant and anti-inflammatory activities [9,77,78]. Most polyphenols cannot be absorbed in their native form and are chemically modified after ingestion. Therefore, knowledge of the bioavailability and metabolism of various polyphenols is necessary to evaluate their biological activities within target tissues. Generally, polyphenols are moderately water-soluble compounds with molecular weights of 300–4000 Da. They share the same phenol carbon ring (5–7 aromatic rings per 1000 Da), albeit with different structures. According to the number of phenol rings and structural elements bound to them, polyphenols are classified into two major groups: flavonoids and non-flavonoids. The most important classes of flavonoids found in foods are flavonols, flavones, flavan-3-ols, anthocyanins, flavanones, and isoflavones. The most important classes of non-flavonoids are phenolic acids, stilbenes, tannins, and lignoids [79,80] (Figure 1).

As discussed in this review, polyphenols exert their beneficial effects on treating cardiovascular diseases, including myocardial infarction. They also exhibit antioxidant, anti-inflammatory, anti-cardiac hypertrophy, anti-atherosclerosis, anti-diabetic, and anti-apoptotic activities through modulating various signaling pathways demonstrated in Figure 2.

Schematic diagram of the involved mechanisms by which polyphenols exert their protective effects. Myocardial infarction and cardiac hypertrophy can be considered mainly with the reduction of NF-ҡB and its molecules. Increased oxidative stress and inflammatory markers are associated mainly in diabetes and atherosclerosis. Polyphenols administration inhibits also the lipogenesis by reducing low density lipoproteins (LDL) cholesterol.

### 3.1. Flavonoids

Flavonoids are an important class of polyphenols that can be found in many kinds of plants, including vegetables and fruits. Table 1 provides examples of these bioactive compounds along with their host foods [81,82,83,84,85,86,87]. The structural diversity between flavonoid subgroups produces different beneficial properties, such as antioxidant, anti-inflammatory, anti-diabetic, and/or antithrombotic activities [88,89,90,91]. In general, more than two-thirds of the polyphenols consumed in one’s diet are flavonoids. To date, more than 4000 flavonoids have been identified and described [92]. Their beneficial effects on the cardiovascular system have been described mainly in relation to the French Paradox phenomenon and the Mediterranean diet. The French Paradox refers to the decreased incidence of coronary heart disease despite a high intake of saturated fat. The Mediterranean diet, rich in fruits and red wine, has also been shown to protect against the development of cardiovascular diseases [93,94]. Flavonoids are widely used as antioxidants due to their phenolic hydroxyl groups, as well as anti-inflammatory agents that act on the diastolic arteries [95]. Chemically, flavonoids have a general structure with a 15-carbon skeleton that consists of two phenyl rings and a heterocyclic ring. This carbon structure can be abbreviated as C6-C3-C6 [96]. The differences between individual flavonoids include the orientation of hydroxylation or methylation, the position of the benzoid substituent, and the degree of unsaturation. Their structural characteristics are the limiting factors for their antioxidant and antiproliferative activities [10]. The most common flavonoids are quercetin, catechin/epicatechin, and myricetin (Table 1).

### 3.2. Non-Flavonoids

Non-flavonoids are found in different kinds of citrus fruits and berries, coffee, olive and sesame oil, cereals, and red wine. The most important class of non-flavonoid polyphenols is phenolic acid. Both in vivo and in vitro studies have shown their antioxidant and anti-inflammatory characteristics due to their ability to reduce the production of inflammatory markers, such as IL-1β, IL-8, MCP-1 (monocyte chemoattractant protein), COX-2, and iNOS [97]. Phenolic acid can protect cells against various injuries, depending on their chemical structures [98]. The main representatives of phenolic acids are derivatives of benzoic or cinnamic acids [99]. These acids include gallic acid, vanillic acid, caffeic acid, and ferulic acid (Table 1). Natural stilbenes are characterized by the presence of a 1,2-diphenylethylene nucleus [100]. They have low bioavailability [101] depending on the route of their administration, absorption, and metabolism. Their metabolism is determined by their chemical structure, molecular size, degree of polymerization, etc. [102,103]. In addition to their antibacterial effects, stilbenes also exert anti-hypertensive and anti-tumor effects, as well as inhibit platelet aggregation [104] and prevent the cell damage induced by oxidative stress, hypoxia, and ischemia [105,106,107,108]. Resveratrol is one of the primary representatives of stilbenes.

## 4. Potential Implications of Polyphenols on MI

A therapeutically relevant strategy for preventing myocardial infarction injury is to restore blood supply to the ischemic area and minimize related damage. As we described above, ischemia/reperfusion injury is a multifactorial complex that involves, among other factors, generating reactive oxygen species and pronounced inflammatory responses. While high amounts of ROS are clearly detrimental, lower amounts of ROS contribute to myocardial conditioning. A variety of enzymatic and non-enzymatic substances have been tested for their antioxidant capacity and, accordingly, their capacity to diminish different cardiovascular diseases, especially myocardial I/R damage. Many synthetic molecules and processes are efficient in preventing CVDs and are thus used for cardioprotection. However, their use is frequently limited due to their side effects [109]. On the other hand, many kinds of natural compounds have been studied for their positive effects on the cardiovascular system without side effects. In recent decades, epidemiological studies and clinical trials have shown associations between the regular intake of natural polyphenols from some plants, vegetables, juices, and red wine, and the reduction of oxidative damage, a process involved in the onset and development of cardiovascular diseases [9,110,111,112]. Consumption of this food may help inhibit the inflammatory process, endothelial dysfunction, and NO production [113], reverse hyperlipidemia, decrease the atherogenicity of LDL particles [114], and protect LDL cholesterol from oxidation [115]. Possible mechanisms include the effects on blood pressure, lipid metabolism, inflammation, endothelial function, oxidative stress biomarkers, and nitric oxide level. The protective role of polyphenols in cardiac ischemia may also be related to their ability to scavenge oxygen free radicals [116], maintain NO concentration [117,118], and inhibit mast cell secretion [119]. Sato et al. [116] found that a non-alcoholic red wine extract could protect the heart from the detrimental effects of ischemia/reperfusion injury by reducing myocardial infarctions in a rat model. Other authors have reported the positive effects of several flavonoids on acute regional myocardial ischemia in isolated rabbit hearts [120]. According to Ning et al. [121], polyphenol administration improved functional recovery in a perfused heart after ischemia through stimulation of the cytochrome P450 system. Cytochrome P450 reductase improves catalytic efficiency and thus diminishes the production of free radicals. Studies with quercetin have revealed another protective pathway by preventing the decrease in the xanthine dehydrogenase to oxidase ratio observed during ischemia/reperfusion in rats [122]. The inhibition of xanthine oxidase by flavonoids has also been described [123]. In addition, polyphenols are found to possess positive chronotropic and antiarrhythmic effects and to minimize mitochondrial ischemia/reperfusion injury [124]. During ischemia/reperfusion, free oxygen radicals accumulate and thus cause a lowering of NO bioavailability. It has been reported that the effects of red wine polyphenols on NO stability and generation are crucial for preventing ischemia [125,126]. Studies at our laboratory described an increase in NO synthase activity due to treatment with red wine polyphenolic compounds in the heart and aorta [127]. Moreover, polyphenols can improve endothelial dysfunction by stimulating the endothelium-derived hyperpolarizing factor [128,129] and increase coronary flow via endothelium-dependent relaxation [130], directly promote NO production in endothelial cells via an increase in eNOS activity due to its phosphorylation [131], and help preserve the integrity of endothelial cells [132]. Notably, resveratrol, a phenolic acid compound, upregulates eNOS activity by increasing the level of eNOS at the gene and protein levels. This is done by increasing eNOS phosphorylation in the serine 1177 residue, by decreasing the endogenous eNOS inhibitor asymmetric dimethylarginine (ADMA) [117], and/or via the AMP-activated protein kinase pathway in the myocardium [98]. Another therapeutically relevant effect of polyphenols is the relaxation of the precontracted smooth muscle of the coronary arteries with intact endothelium. Some studies have also observed the relaxation of the endothelium-denuded arteries [133]. According to several authors, the induction of endothelium-dependent relaxation by polyphenols could occur via the enhanced generation and/or increased biological activity of NO [134,135]. In the following section, we have summarized the cardioprotective effects of several naturally occurring polyphenols.

### 4.1. Effects of Resveratrol

Resveratrol (*E*-3,5,4′-trihydroxystilbene) is a natural polyphenolic compound, mainly found in edible plants, such as peanut, grape, berry, and red wine. Many studies have shown that resveratrol affects a number of key cellular pathways and molecular targets with a wide range of biological effects. For example, resveratrol attenuates the pathological progression in a variety of disease models, such as diabetes mellitus, cancer, obesity, and neurodegenerative disease [136]. The biological and pharmacological properties of resveratrol have been well established, including its antioxidant and anti-inflammatory activities, anti-mitochondrial dysfunction, and anti-apoptotic potency. It has strong biological activity, which can also help prevent free radicals, oxidative stress, and tumor diseases as well. Metabolic modulation and angiogenesis are also identified as therapeutic actions of resveratrol. Among its other therapeutic actions, resveratrol has been reported as a promising cardioprotective compound against ischemic heart disease in vivo, especially myocardial I/R injury, by modulating angiogenesis, oxidative stress, inflammation, cardiomyocyte apoptosis, and mitochondrial functions, as well as energy metabolism [107,137,138,139]. An experimental study has identified the different molecules and pathways involved in the protective effects against ischemia/reperfusion injury. The first possible mechanism described in the literature indicates that the Nrf2/ARE pathway plays a principal role in the protective effect against apoptosis and oxidative stress [138]. In the same study, the protective effect of resveratrol on I/R injury has been described by the reduced infarct area, reduced creatine-kinase level, and increased glutathione peroxidase and superoxide dismutase biosynthesis. Another group has reported that the TLR4/NF-κB signaling pathway is also involved in the protective effects of resveratrol after I/R injury [139]. In this study, resveratrol suppressed free radical generation. TLR4 is rapidly upregulated to mediate proinflammatory cytokines in response to I/R injury, which conspires to induce myocardial damage [107]. Subsequently, TLR4 predominantly activates the NF-κB family, the key transcription factors in modulating the inflammatory response, as well as cell death genes linked to the pathogenesis of the cardiovascular disease. Surprisingly, resveratrol treatment decreases the expression of TLR4 and NF-κB in the reperfused myocardium, accompanied by lower levels of tumor necrosis factor-α (TNF-α) and reduced infarct size. Resveratrol is a strong activator of SIRT1 and is potentially anti-atherosclerotic [140]. An experimental study by Dong et al. showed that resveratrol protected the myocardium against I/R damage through deactivation of the pyrin domain-containing-3 (NALP3) inflammasome and suppression of the IL-1β- and IL-18-mediated inflammatory cascade [141]. Resveratrol also protects the heart from fibrotic remodeling and cardiac hypertrophy. Another proposed mechanism includes the inhibition of cytokine release and the modulation of nitric oxide, which can result in antioxidant and anti-inflammatory action [142]. Moreover, the resveratrol-related improvement of coronary flow during reperfusion is associated with increased eNOS, pAkt, and SIRT1 protein expression in Goto–Kakizaki rat hearts involved in the cardioprotection against I/R. Furthermore, resveratrol prevents the heart against remodeling and hypertrophy [143].

Experimental studies have shown that resveratrol has positive effects on cardiovascular and other diseases. Unexpectedly, there has been very little success in transferring resveratrol into clinical practice with the desired efficacy in relevant patients despite its promising infarct-limiting effects in animal studies. Theoretically, animal models help to explore the probable mechanisms of a drug; however, the anatomic and/or physiological differences between different species can lead to inconsistencies between preclinical studies and clinical studies.

Many clinical studies have investigated the effects of resveratrol intake in the context of cardiovascular diseases. The results depend on the amount of resveratrol used (from 5 to 5000 mg/day) and the duration of treatment (from days to months). Most studies have described the positive effects of resveratrol-treatment on systolic blood pressure, total cholesterol level, and fasting glucose at a dose of more than 300 mg of resveratrol per day [144,145]. Some clinical studies have shown the reduced expression of pro-inflammatory cytokines, such as chemokine CCL3, IL-1β, and TNF-α, as well as modified patterns of inflammatory-related microRNAs in peripheral blood mononuclear cells [141,146] due to the resveratrol-enriched grape extract.

### 4.2. Effects of Catechin/Epicatechin and Its Derivates

Catechins are flavan-3-ols found mainly in dark chocolate, tea, and nuts [147]. Experimental studies have shown catechins to exert a variety of effects on the cardiovascular system [148], including vasodilatation, the inhibition of platelet adhesion, inflammation, reactive oxygen species generation, low-density lipoprotein oxidation, and the activation of endothelial nitric oxide synthase [149]. Regular chocolate intake has beneficial effects on arterial stiffness [150]. Catechins play an important role in the prevention of vascular dysfunction, mainly through the elimination of free radicals in order to prevent lipid oxidation [151]. Meta-analyses have reported that high flavon-3-ols intake can reduce the risk of myocardial infarction [152]. Some authors have described possible therapeutically relevant strategies using catechin for the treatment of ischemia/reperfusion injury, showing a reduced infarct size in rats treated with epicatechin 2 or 10 days prior to I/R injury. These studies have also observed decreased ROS generation, thereby improving the coronary flow and nitric oxide metabolite level [153]. Numerous epigallocatechin-3-gallate studies have described similar therapeutic effects, mainly associated with the inhibition of LDL cholesterol, NF-κB, and myeloperoxidase activity, a reduction in inflammatory markers, and ROS inhibition. Epigallocatechin-3-gallate (EGCG) inhibits apoptosis, reduces chemokine-induced neutrophil chemotaxis in vivo, and also reduces inflammatory responses and fibrosis. EGCG has also been shown to reduce the infarcted area in a diabetic model of I/R injury [154]. Another study has highlighted the protective effects of EGCG on myocardial I/R injury via the inhibition of apoptosis targeting the PI3/Akt signaling pathway, which has also been shown to improve dilated cardiac remodeling with reduced cardiac contractility and a decreased iNOS expression level [155]. Some authors have demonstrated its positive effects on reducing the IL-6 plasma concentration, myeloperoxidase activity (MPO), the creatine phosphokinase (CPK) concentration, and the NF-κB expression level in rats with experimentally induced MI, along with inhibition of the pro-inflammatory cytokine TNF-α level in the serum [156]. EGCG also maintains the redox balance by upregulating SOD and CAT activities while limiting lipid peroxidation. In addition, EGCG administration can ameliorate apoptotic markers, such as Bax and caspase 3 and 9, accompanied by the inhibition of DNA fragmentation and the downregulation of p53. EGCG treatment is able to attenuate MI by reducing the infarct size in Langendorff perfused rat hearts by acting on the adenosine receptor (ADR) and opioid receptor (OPR). Other scientists have demonstrated that a combination of EGCG (10 μM) with Zn^2+^ (5 μM) can enhance anti-apoptotic activity and protect H9c2 cells through activation of the PI3K/Akt pathway. In addition, downregulation of the expression levels of TNF-, IL-6, and IL-8 has been observed, suggesting EGCG can be applied to the prevention of MI in clinical practice. Similar results have been shown after -(-) epicatechin treatment, resulting in a significant reduction of the infarct size, which was sustained up to 3 weeks after injury, accompanied by preserved myocardial inflammation, as well as decreases in matrix metalloproteinase activity and tissue oxidative stress. Moreover, activation of the pro-survival protein kinase AKT and ERK ½ pathways has been demonstrated to confer cardioprotection [45]. Another study has also shown that a catechin derivate inhibits inflammation by suppressing TLR4/NF-κB signaling [157].

### 4.3. Effects of Quercetin

Quercetin (3,3′,4′,5,7-pentahydroxyflavone) is a common flavonoid highly enriched in fruits (mainly berries and vegetables). Its molecule has a lipophilic character, despite the presence of five hydroxyl groups. These groups are responsible for quercetin’s reactivity and its biological ability to bind different kinds of substituents and thus create derivates. Experimental studies have shown its role in cardiovascular, nervous, gastrointestinal, and other systems, as well as its ability to scavenge free radicals by chelating Fe^2+^ and Cu^+^ ions [158]. It has been found that acute quercetin administration 15 min prior to the onset of ischemia or during reperfusion can lead to an improvement in the functional parameters of the heart, such as left ventricular pressure (LVDP), contractility parameters (+(dP/dt) max) and (−(dP/dt) max), as well as produce a significantly lower increase in end-diastolic pressure [159]. Similarly, a beneficial effect of quercetin has been observed after its acute administration directly during the reperfusion of isolated rat hearts (I/R 30 min/30 min) by inhibiting HMGB1 (high mobility group box 1) [160]. Another study has shown the beneficial effects of quercetin on the ultrastructure of the left ventricle by preventing the activation of MMP-2, the metalloproteinase involved in remodeling the extracellular matrix; also, its increased SOD expression has prevented the induction of apoptosis [161]. Reduced apoptosis after quercetin treatment has also been demonstrated as a cardioprotective effect due to decreased Bax expression and simultaneously increased Bcl-2 protein expression [162]. The same result has been confirmed in rats with chronically administered quercetin, as well as on cells of the myoblast cell line H9c2 [163]. The positive effect of quercetin against pathological levels of ROS has been confirmed by the reduced expression levels of eNOS, iNOS, and NOX2 (NADPH oxidase) [164]. Direct evidence of the antioxidant effect of quercetin has been demonstrated by the suppression of lipid peroxidation over long-term quercetin administration prior to myocardial infarction in a rat model. The cardioprotective potential of quercetin has also been confirmed against experimentally induced autoimmune myocarditis progression by suppressing oxidative stress via endothelin-1/MAPK (mitogen-activated kinase compound) signaling. Quercetin induces an increase in Akt phosphorylation as part of the PI3K/Akt (phosphatidylinositol-3-kinase/Akt) signaling pathway, which is associated with its cardioprotective effect against I/R [161]. The PI3K/Akt signaling pathway is involved in the modulation of cell proliferation, survival, and apoptosis. An increase in Akt phosphorylation has also been demonstrated with quercetin administration during reperfusion only. It is assumed that quercetin may not increase cell viability by activating the PI3K/Akt survival pathway only but also by reducing apoptosis via the inhibition of c-Jun N-terminal kinases (JNK) and p38 kinase phosphorylation, which directly or indirectly inhibits caspase-3 and Bax protein activation in the I/R model in H9c2 cells. It has been reported that quercetin exerts anti-inflammatory activity by inducing the expression of heme oxygenase-1 (HO-1). This mechanism is associated with the activation of the Keap1-Nrf2 signaling pathway [164]. They have described the anti-inflammatory, antioxidant, and anti-apoptotic effects of quercetin in rats under I/R conditions via decreased superoxide dismutase and catalase activity, as well as via a reduction of apoptosis, thereby protecting against myocardial damage. During ischemia, primary cytokines, such as IL-1 and TNF-α (along with an increase in the concentration of C-reactive protein (CRP)), are released into the bloodstream. In this context, chronically administered quercetin has been demonstrated to reduce the levels of the pro-inflammatory markers IL-1, TNF-α, and CRP compared to a control group. The effect of quercetin has also been confirmed in two other studies where it suppresses TNF-α while reducing pro-inflammatory cytokine IL-6 levels. However, it also increases anti-inflammatory cytokine IL-10 levels [165]. In another animal study, quercetin has protected isolated hearts against I/R injury by inhibiting the inflammatory markers TNF-α and IL-1β [166]. Moreover, quercetin has protected the myocardium by suppressing TLR4 and NF-κB protein expression, thus playing a role in modulating inflammation and oxidative stress by regulating the TLR4/NF-κB pathway [167].

## 5. Conclusions

Polyphenol-rich food consumption has been associated with many health benefits, including the amelioration of cardiovascular problems, and represents an important source of novel drugs. Polyphenols have multiple molecular targets and act on different protective pathways. Experimental in vitro and in vivo studies and a few clinical trials have demonstrated the effectivity of polyphenol compounds as an antioxidant, anti-inflammatory, and anticancer drugs. Despite numerous experimental studies, there is a great clinical need to enhance myocardial recovery in patients with myocardial infarctions. Unlike drugs, different kinds of polyphenols affect several key metabolic pathways. Elucidation of the protective effect of polyphenols against myocardial infarction may improve the recovery of the heart’s function and structure and thus prolong the patient’s lifespan. Natural products thus have significant potential for the treatment of cardiovascular diseases and, hopefully, will offer great value in future applications.

## Figures and Tables

**Figure 1 molecules-25-03469-f001:**
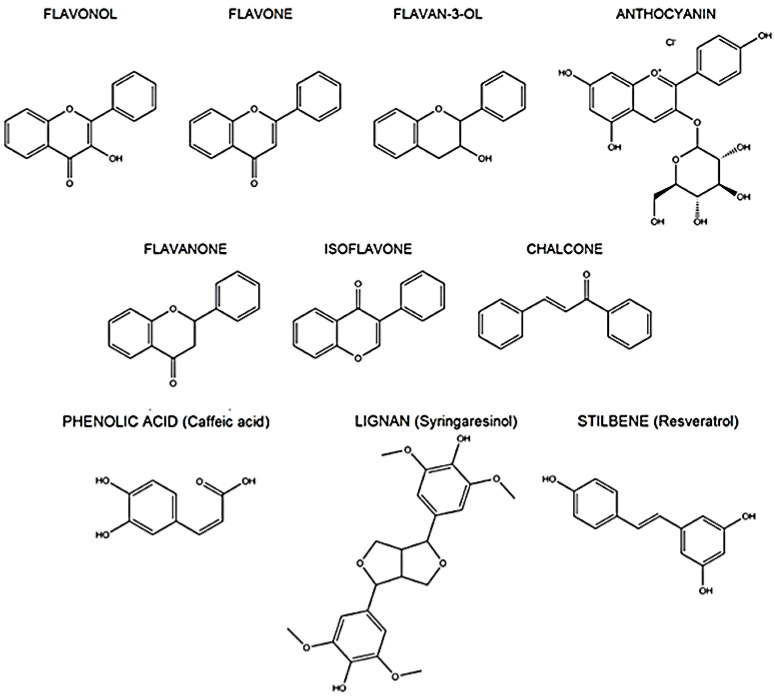
Chemical structure of the most important polyphenols.

**Figure 2 molecules-25-03469-f002:**
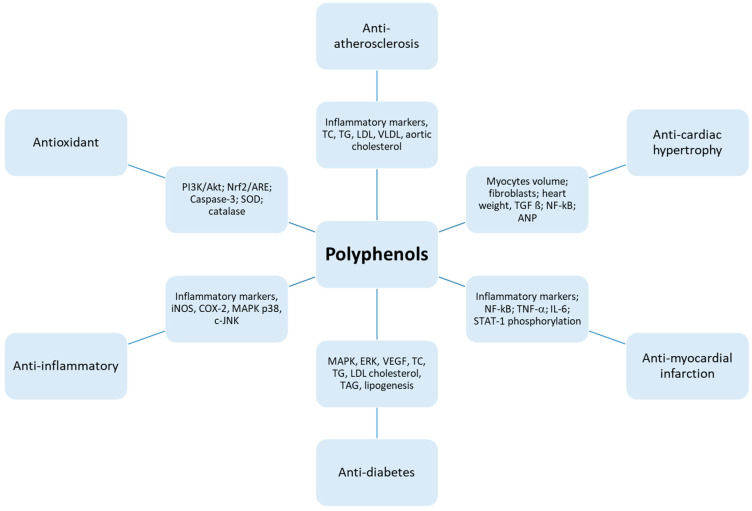
Depiction of the mechanistic profile of polyphenols.

**Table 1 molecules-25-03469-t001:** Food sources of polyphenol subclasses.

Subclasses	Examples	Food Sources
Flavonols	Kaempferol	kale, beans, tea, spinach, broccoli
	Myricetin	nuts, berries, tea, red wine
	Quercetin	red onion, kale, leaves, seeds, fruits, grains
	Rhamnetin	cloves
Flavones	Apigenin	parsley, celery, chamomile
	Hispidulin	plants (Grindelia, Saussurea, Salvia)
	Luteolin	barks, rinds, leaves
	Tangeretin	tangerine, citrus peels
Flavan-3-ols	Catechin	cocoa, prune juice, broad bean, peach, green tea
	Epicatechin	dark chocolate, blueberries, green tea
	Gallocatechin	green, white and black tea
	Epigallocatechin gallate	green, white and black tea
Flavanones	Butin	seeds
	Eriodictyol	lemon, herbs
	Hesperidin	citrus fruits
	Naringenin	grapefruit, herbs
Anthocyanins	Cyanidin	blueberries, cranberries
	Delphinidin	eggplant, different kinds of berries
	Malvidin	red wine, colored fruits, the skin of red grapes
	Peonidin	cranberries, plums, grapes
Isoflavones	Daidzein	soybeans, black beans, green
	Formononetin	soybeans
	Genistein	soybeans
	Glycitein	soybeans
Phenolic acid	Gallic acid	Aronia, green tea, grape seeds
	Caffeic acid	coffee beans, green tea,
	Ferulic acid	cereals, corn, rice
	Vanillic acid	Aronia, green tea, berries
Stilbenes	Resveratrol	skin of grapes, red wine
Lignoids	Pinoresinol	sesame seed, olive oil
	Sesamin	sesame oil

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
