# Peer review of "Protective Effects of Polyphenols against Ischemia/Reperfusion Injury"

_molecules, 2020, doi:10.3390/molecules25153469_

Round 1

Reviewer 1 Report

This manuscript written by Martina Cebova and Olga Pechanova described the protective effects of polyphenols against ischemia/reperfusion injury. The authors spent a lot of efforts to describe the type of phenolic compounds. However, those are well-known information. The most valuable part of this paper is section 4. However, the myocardial infarction related data of quercetin, catechin/epicatechin, and resveratrol are barely addressed. 

1. The benefit (antioxidant, anti-inflammatory) and sources (table 1) of phenolic compounds are well-known. However, the authors spend a lot of paragraphs on it.

2. The myocardial infarction related data of quercetin, catechin/epicatechin, and resveratrol are barely addressed.

3. There is no SAR part of active compound, which cannot really provide useful data for readers.

Therefore, I feel this paper cannot be accepted and published in Molecules.

Author Response

We would like to thanks reviewer for reading our manuscript. We highly appreciate the reviewer`s interest in our research, favorable comments and constructive suggestions. Since our manuscript was revised by English editing office, it was difficult to highlight the changes in yellow. We incorporated all changes in the manuscript.

Comments and Suggestions for Authors

This manuscript written by Martina Cebova and Olga Pechanova described the protective effects of polyphenols against ischemia/reperfusion injury. The authors spent a lot of efforts to describe the type of phenolic compounds. However, those are well-known information. The most valuable part of this paper is section 4. However, the myocardial infarction related data of quercetin, catechin/epicatechin, and resveratrol are barely addressed. 

  1. The benefit (antioxidant, anti-inflammatory) and sources (table 1) of phenolic compounds are well-known. However, the authors spend a lot of paragraphs on it.

We agree that anti-oxidant and anti-inflammatory benefits of polyphenols are known. However, we tried to concentrate only on their effects on cardiovascular injury focusing on treatment of myocardial infarction in animal and human studies. To our best knowledge, this will be the first review to be focused on it from this point of view.

  1. The myocardial infarction related data of quercetin, catechin/epicatechin, and resveratrol are barely addressed.

The manuscript was revised and some other mechanisms were involved into the text.  We added another pathomechanism of myocardial infarction related to quercetin, catechin/epicatechin and resveratrol treatment. We rephrased and reformatted some paragraphs as well as we added an illustrative figures showing depiction of mechanistic profile of polyphenols on cardiovascular system.

  1. There is no SAR part of active compound, which cannot really provide useful data for readers.

In each reported polyphenol we mentioned the structure of compound. Exact groups responsible for reactivity and its biological activity were added into the manuscript.

Reviewer 2 Report

The paper reviewed the effects and benefits of polyphenols on cardiovascular injury focusing on treatment of myocardial infarction in animal and human studies. The polyphenol compounds have been studied for their pharmacological properties such as antioxidant, anti-inflammatory effects and free radical scavenging and proven for treatment of cardiovascular diseases. The introduction (items 1 and 2) is particularly interesting and presents the various aspects involved in the disease. However, some corrections listed below should be done :

Item 3, especifically in itens 3.1. and 3.2.: authors described general characteristics of polyphenols. In literature, various papers describe the antioxidant, free radical scavenging and anti-inflammatory effects of both classes of polyphenol compounds. However, the authros do not cited these works. Therefore, my suggestion is : some literature papers reporting these properties to flavonoids, caffeoylquinic acids and stilbenoids must be cited.

Lines 222-223: Authors wrote: "Generally, polyphenols are moderately water-soluble compounds with molecular weight of 500–4000 Da". The range of molecular weight is very wide. Additionally, several polyphenols such as the compounds quercetin, luteolin and quinic acid derivatives have molecular weights below 500. Therefore, adjustments should be done.

Line 228: I have two suggestions for authors: tannins must be included among the examples of non-flavonoid class. The term lignans should be modified for lignoids, because lignans are a subclass of lignoids.

In figure 1, the examples of polyphenol compounds show structures of substances for non-flavonoid class and skeletons pertaining to flavonoid class. Authors should standardize the figure and to present only examples of compounds, supressing the skeleton types.

The names kaempferol, rhamnetin and epigallocatechin gallate must be corrected in Table 1.

Line 317: change the stereochemical prefix trans- to E-. The groups bonded to sp2 carbons are not identical, therefore the correct designation is E.

Line 375: there is a reference (Larson et al., 2016) erroneously cited in the text.

Line 390: the charge of anion must be superscripted.

Author Response

We would like to thanks reviewer for reading our manuscript. We highly appreciate the reviewer`s interest in our research, favorable comments and constructive suggestions. Since our manuscript was revised by English editing office, changes are not highlighted in yellow.

Comments and Suggestions for Authors

The paper reviewed the effects and benefits of polyphenols on cardiovascular injury focusing on treatment of myocardial infarction in animal and human studies. The polyphenol compounds have been studied for their pharmacological properties such as antioxidant, anti-inflammatory effects and free radical scavenging and proven for treatment of cardiovascular diseases. The introduction (items 1 and 2) is particularly interesting and presents the various aspects involved in the disease. However, some corrections listed below should be done :

Item 3, especifically in itens 3.1. and 3.2.: authors described general characteristics of polyphenols. In literature, various papers describe the antioxidant, free radical scavenging and anti-inflammatory effects of both classes of polyphenol compounds. However, the authros do not cited these works. Therefore, my suggestion is : some literature papers reporting these properties to flavonoids, caffeoylquinic acids and stilbenoids must be cited.

Answer: we added citations according reviewer`s suggestion.

Lines 222-223: Authors wrote: "Generally, polyphenols are moderately water-soluble compounds with molecular weight of 500–4000 Da". The range of molecular weight is very wide. Additionally, several polyphenols such as the compounds quercetin, luteolin and quinic acid derivatives have molecular weights below 500. Therefore, adjustments should be done.

Answer: We adjusted value of molecular weight according to reviewer`s suggestion.

Line 228: I have two suggestions for authors: tannins must be included among the examples of non-flavonoid class. The term lignans should be modified for lignoids, because lignans are a subclass of lignoids.

Answer: We added tannins between examples of non-flavonoids class as well as we modified term lignoids instead of lignans according reviewer`s suggestion.

In figure 1, the examples of polyphenol compounds show structures of substances for non-flavonoid class and skeletons pertaining to flavonoid class. Authors should standardize the figure and to present only examples of compounds, supressing the skeleton types.

Answer: We standardized the figure 1 according the reviewer`s suggestion.

The names kaempferol, rhamnetin and epigallocatechin gallate must be corrected in Table 1.

Answer: Spelling mistakes were corrected, we apologized for the mistakes.

Line 317: change the stereochemical prefix trans- to E-. The groups bonded to sp2 carbons are not identical, therefore the correct designation is E.

Answer: Stereochemical prefix was corrected accrding reviewer`s suggestion.

Line 375: there is a reference (Larson et al., 2016) erroneously cited in the text.

Answer: Name of the author were deleted, we apologize for the mistake.

Line 390: the charge of anion must be superscripted.

Answer: All charges either anions and cations are supercribed. 

Reviewer 3 Report

REVIEWER’S COMMENTS

for:

Manuscript entitled “Protective Effects of Polyphenols Against Ischemia/Reperfusion Injury”

Authors: Martina Cebova, Olga Pechanova

Journal: MDPI Molecules

Summary: In this current review, the authors demonstrate the potential cardioprotective effects of polyphenols against acute myocardial infarction. The writers emphasize the anti-inflammatory and antioxidant effects of this compounds and detail the impact of resveratrol, catechin/epicatechin and quercetin in the ischemia/reperfusion injury.

The topic is actual and interesting as the cadiovascular diseases are responsible for 17,9 million people death every year (WHO data). In the current scientific literature, there are numerous articles which demonstrate the cardioprotective effects of polyphenols but there are only a few which emphasize the effectivity of polyphenols specificly against acute myocardial infarction (MI). Despite this review has been written based on 152 articles, it demonstrates only exiguous number of signal transduction pathways and doesn’t detail the exact pathomechanism of MI. Although this writing emphasizes the role of ischemia/reperfusion (I/R) injury in the pathophysiology of MI, the lipid profile improving, anti-thrombotic, anti-atherogenic and antihypertensive effects of polyphenols are also crucial in the prevention of heart attack. According to this reviewer’s opinion, this review should enumerate more than 4 cardioprotective compounds and confirm the effectivity of this agents by concrete examples about animal/clinical experiments.

I have the following specific comments:

Major comments

  1. Regarding this writing is a review, the authors should have demonstrated more molecular pathways which have key role in the pathomechanism of myocardial infarction – E.g. SIRT1, COX, PPARγ etc.
  2. There was a great amount of repetitions in the text, and a number of sentences didn’t reveal new information.
  3. Illustrative figures are missing from this current paper. As this reviewer see it, using figures could facilitate the understanding, e.g. illustrations about the hypothetic molecular mechanisms of polyphenols.

Minor comments

  1. There are numerous linguistic errors. E.g.: line 66 – instead of “with the” the writers should use “resulting” because the Na overload is a consequence; line 71 – the expression “main source” is incorrect, the authors should use e.g. “main signal transduction pathway”; line 73 - the writers used the apoptosis word twice within a sentence.; line 96. - instead of “supportive tissue”, the “connective tissue” expression is more acceptable; line 107 - the word “establishment” is incorrect in this context; line 201. - the correct form is: “Massive alteration in intracellular Ca2+ level…”; line 301. - incorrect formulation (“level of eNOS on gene”), e.g. “increased/raised eNOS gene expression” would be preferable; line 331. - the mechanism should be used in plural; line 324. - incorrect expression (“prevent free radicals”) it would be more acceptable to use e.g. “prevent free radical formation/production”.; line 346. - a “h” is missing in the end of “which”; line 370. - the verb is missing from the sentence; line 375. - “desribed posible” is incorrect
  2. Line 26 – Acut myocardial infarction is the most severe complication of coronary artery disease. The sentence which describe the prevalence of CV deaths is incomprehensible.
  3. Line 29 – By the listing of risk factors the authors missed to mention one of the most important life threating conditions, the obesity.
  4. Line 41 – The writers could apply a more sophisticated expression for the description of MI. (eg. Acute myocardial infarction is a potentially life-threatening condition)
  5. Line 156.: The authors have used “heart failure” as the synonym of myocardial infarction. Not every type if heart failures have inflammatory background and for this reason this sentence is incorrect.
  6. Line 169: The authors should concretize on which cell types has been the TLR4 expression investigated in the peripheral blood
  7. Line 88, line 110, line 200, line 241, line 260, line 399 - those sentences need to be rephrased, because they are either grammatically or logically incorrect.
  8. The role of the NO and NOS is not clear in this review. The authors state that all isoforms of NOS are activated due to ischemia (line 151) however later (line 301.) an increased NOS activity meant to be beneficial after red wine polyphenol therapy.
  9.  

The scientific English of the review is of moderate quality, the numerous errors (both grammatical and syntax) need correction, and a proofreading by a native English-speaker is recommended. In summary, the general idea to write a review about polyphenols which can mitigate the I/R injury after acute myocardial infarction is adequate, for this reason, this reviewer encourages the authors to rephrase and reformat the writing and complete it with further compounds, molecular mechanisms and figures.

Author Response

We would like to thanks reviewer for reading our manuscript. We highly appreciate the reviewer`s interest in our research, favorable comments and constructive suggestions. Since our manuscript was revised by English editing office, it was difficult to highlight the changes in yellow. We incorporated all changes in the manuscript.

Major comments

  1. Regarding this writing is a review, the authors should have demonstrated more molecular pathways which have key role in the pathomechanism of myocardial infarction – E.g. SIRT1, COX, PPARγ etc.

Answer: According to reviewer`s suggestions we added another pathomechanism of myocardial infarction.

The cyclooxygenase (COX) enzyme pathway is involved in almost all aspects of health and disease, including myocardial infarction. COX has two isoforms: COX‐1 is constitutively expressed in many cell types, whereas COX‐2 is constitutively expressed only in certain regions but is readily induced in inflammation and cancer [56]. To date, some studies have shown the cross-communication between iNOS and the COX-2 enzyme, suggesting a key link between these enzymes in pathological conditions. During inflammation, the NO produced by iNOS contributes to the upregulation of COX-2 and thus increases its cytotoxic effects. Additionally, iNOS is also known to be co-inducted with COX-2. Both COX-2 and iNOS may contribute to the development and progression of coronary arterial disease [57]. Some other studies showed that peroxisome proliferator-activated receptor gamma (PPAR-γ) ligands also reduce the tissue injury associated with myocardial I/R via inhibition of the JNK/AP-1 pathway [58]. Moreover, administration of a PPAR-γ agonist reduced the myocardial infarction size by up to 60% along with the release of creatinine kinase, as well as an improvement in cardiac function [59].

SIRT 1 (Silent mating-type information regulation 1) appears to have a prominent role in cardiovascular biology; in preclinical models, it promotes a variety of physiological effects. It has been shown, for example, to protect the heart against I/R injury, hypertrophy, and cardiomyocyte apoptosis. SIRT1 overexpression leads to reduced myocardial hypertrophy, interstitial fibrosis, and oxidative stress [75]. These changes were associated with a significant improvement in cardiac function, as assessed by the ejection fraction and fractional shortening. Moreover, SIRT1 ameliorates endothelial function via the activation of eNOS and prevents macrophage foam cell formation and the calcification of vascular smooth muscle cells. Subsequently, studies on genetically engineered mouse models demonstrated that Sirt1 exerts atheroprotective effects by activating eNOS or diminishing the NF-κB activity in endothelial cells and macrophages. Another report showed the protective role of Sirt1 in vascular smooth muscle cells against DNA damage, medial degeneration, and atherosclerosis [76].

  1. There was a great amount of repetitions in the text, and a number of sentences didn’t reveal new information.

Answer: We corrected the text trying to not repeat the information.

  1. Illustrative figures are missing from this current paper. As this reviewer see it, using figures could facilitate the understanding, e.g. illustrations about the hypothetic molecular mechanisms of polyphenols.

Answer: We added an ilustrative figure showing depiction of mechanistic profile of polyphenols on cardiovascular system.

Minor comments

  1. There are numerous linguistic errors. E.g.: line 66 – instead of “with the” the writers should use “resulting” because the Na overload is a consequence; line 71 – the expression “main source” is incorrect, the authors should use e.g. “main signal transduction pathway”; line 73 - the writers used the apoptosis word twice within a sentence.; line 96. - instead of “supportive tissue”, the “connective tissue” expression is more acceptable; line 107 - the word “establishment” is incorrect in this context; line 201. - the correct form is: “Massive alteration in intracellular Ca2+ level…”; line 301. - incorrect formulation (“level of eNOS on gene”), e.g. “increased/raised eNOS gene expression” would be preferable; line 331. - the mechanism should be used in plural; line 324. - incorrect expression (“prevent free radicals”) it would be more acceptable to use e.g. “prevent free radical formation/production”.; line 346. - a “h” is missing in the end of “which”; line 370. - the verb is missing from the sentence; line 375. - “desribed posible” is incorrect

Answer: Our manuscript was revised by native speaker, therefore we hope all linguistic errors were corrected. We apologize for mistakes and incorrect describtion.

  1. Line 26 – Acut myocardial infarction is the most severe complication of coronary artery disease. The sentence which describe the prevalence of CV deaths is incomprehensible.

Answer: We corrected the sentence:

Of total cardiovascular deaths, coronary artery disease is the leading cause at 60%, while acute myocardial infarction (MI) comprises 25%

  1. Line 29 – By the listing of risk factors the authors missed to mention one of the most important life threating conditions, the obesity.

Answer: We added obesity as an important life condition.

  1. Line 41 – The writers could apply a more sophisticated expression for the description of MI. (eg. Acute myocardial infarction is a potentially life-threatening condition)

Answer: We described MI more sophisticated.

  1. Line 156.: The authors have used “heart failure” as the synonym of myocardial infarction. Not every type if heart failures have inflammatory background and for this reason this sentence is incorrect.

Answer: We apologize for using incorrect words. We corrected the sentence: TLRs (Toll like receptors) have been shown to participate in the process of myocardial infarction as innate immune factors.

  1. Line 169: The authors should concretize on which cell types has been the TLR4 expression investigated in the peripheral blood

Answer: We added cell tye in the sentence. Moreover, clinical studies have shown that TLR4 expression in peripheral blood mononuclear cells is extremely elevated in patients with myocardial infarction and even higher at the site of the plaque rupture.

  1. Line 88, line 110, line 200, line 241, line 260, line 399 - those sentences need to be rephrased, because they are either grammatically or logically incorrect.

Answer: All sentences mentioned above were grammatically and logically corrected. We apologize for the mistakes.

  1. The role of the NO and NOS is not clear in this review. The authors state that all isoforms of NOS are activated due to ischemia (line 151) however later (line 301.) an increased NOS activity meant to be beneficial after red wine polyphenol therapy.

Answer: We desribed the role of the NO and NOS more clearly. We apologize for the mistakes.

The scientific English of the review is of moderate quality, the numerous errors (both grammatical and syntax) need correction, and a proofreading by a native English-speaker is recommended. In summary, the general idea to write a review about polyphenols which can mitigate the I/R injury after acute myocardial infarction is adequate, for this reason, this reviewer encourages the authors to rephrase and reformat the writing and complete it with further compounds, molecular mechanisms and figures.

Answer: The manuscript was checked and corrected by MDPI English editor, as well as figure and other mechanisms playing role in myocardial infarction were added. 

Round 2

Reviewer 1 Report

This manuscript written by Martina Cebova1 and Olga Pechanova described the protective effects of polyphenols against ischemia/reperfusion injury. This manuscript was revised according to several reviewer’s comments. The myocardial infarction related data of quercetin, catechin/epicatechin, and resveratrol are addressed. However, I feel the novelty of this paper is low. The authors overexpressed the antioxidant, anti-inflammatory, and anticancer for the treatment of myocardial infarction. It is not practical to treat myocardial infarction by those natural products.

Author Response

Dear reviewer,

We would like to thank you for reading our manuscript again. We agree that there are papers oriented on the effect of polyphenolic compounds on cardiovascular system. However, to our best knowledge, there is no paper particularly describing the effect of resveratrol, catechin/epicatechin and quercetin on myocardial infarction consequences and ischemia/reperfusion injury mechanisms. Since there are clinical trials active, recruiting and already completed investigated the effects of polyphenol intake, we assume that polyphenols `treatment may help to protect the myocardium after infarction. Therefore, review has been focused on their various bioactivities such as antioxidant, anti-inflammatory effects and free radical scavenging and we summarized the effect and benefits of polyphenols on cardiovascular injury, particularly on treatment of myocardial infarction in animal and human studies.

Reviewer 2 Report

Dear Authors,

I received the version with English corrections, i.e. with all modifications from English editing office.  English language was improved. However, some corrections suggested by this reviewer were done by authors and excluded by the editing office, such as:

Item 3, especifically in itens 3.1. and 3.2.: authors described general characteristics of polyphenols. In literature, various papers describe the antioxidant, free radical scavenging and anti-inflammatory effects of both classes of polyphenol compounds. However, the authors do not cited these works. Therefore, my suggestion is : some literature papers reporting these properties to flavonoids, caffeoylquinic acids and stilbenoids must be cited.

Lines 222-223: Authors wrote: "Generally, polyphenols are moderately water-soluble compounds with molecular weight of 500–4000 Da". The range of molecular weight is very wide. Additionally, several polyphenols such as the compounds quercetin, luteolin and quinic acid derivatives have molecular weights below 500. Therefore, adjustments should be done.

Line 228: I have two suggestions for authors: tannins must be included among the examples of non-flavonoid class. The term lignans should be modified for lignoids, because lignans are a subclass of lignoids.

The names kaempferol, rhamnetin and epigallocatechin gallate must be corrected in Table 1.

Line 317: change the stereochemical prefix trans- to E-. The groups bonded to sp2 carbons are not identical, therefore the correct designation is E.

I really don't understand why I have to indicate the same corrections twice. There are technical corrections that must be done !!! If the editing office doesn't know them ...... I am so sorry!

Author Response

Dear reviewer,

We would like to thank you for reading our manuscript again. We are really sorry that the latest version that was resubmitted appears to be an English pre-edited version as well as our correction and added parts were omitted. We double check all your comments and points and edit the latest version in the system.

Reviewer 3 Report

Manuscript entitled “Protective Effects of Polyphenols Against Ischemia/Reperfusion Injury”

Authors: Martina Cebova, OlgaT Pechanova

Journal: MDPI Molecules

According to this reviewer’s opinion, after major revision, the authors have made notable changes in this review. As a consequence, this writing became more coherent, more logical and embrace more information about acute myocardial infarction and ischemia/reperfusion injury. This reviewer appreciates the figure about the hypothetic effects of polyphenols although the explanation under the figures are missing. The scientific English of the review is still of moderate quality, the numerous errors (both grammatical and syntax) need correction, and a proofreading by a native English-speaker is recommended.

I have the following specific comments:

Major comments:

  1. Line 28 – Acut myocardial infarction is the most severe complication of coronary artery disease. In this review there are two different prevalence for AMI and CAD, which is incorrect.
  2. The explanation under the figures are missing (except of figure 2., but the text under the picture is crossed out)
  3. The NO can have either beneficial and detrimental biological effects. The authors state that all isoforms of NOS are activated due to ischemia, but later there is a statement which declare that NO level is decreased in early stage of reperfusion. During the reperfusion period the oxidative stress has a pivotal role in the IR injury, and higher NO may have crucial role this process with the formation of peroxynitrate. Taking everything into account, the role of NO in myocardial infarction and IR injury is still not clear in this review.

Minor comments:

  1. There is still a numerous linguistic error. E.g.: line 54. – “…caused by the formation…”; line 58 – the word “mechanism” should be used in plural; line 59 – the word “cardiomyocyte” should be used in plural; line 69 – “The intracellular concentration of sodium ions is significantly increased and it/this phenomenon/this process activates the Na+-H+ ion exchanger – without a noun the sentence is not correct.; line 73. - instead of “with” the writers should use “resulting”/”as a consequence”/”as a result” because the Na overload is a consequence of poor 3Na+-2K+ ATPase function; line 75 - the expression “main source” is incorrect again, the authors should use e.g. “main signal transduction pathway” etc.; line 78. – “elevates”; line 107. – the word “pathway” is superfluous in the end of the sentence; line 112. - the word “establishment” is incorrect in this context, “formation”, “generation” would be preferable; line 118 – the word “tissue” is crossed out in the text notwithstanding that it is necessary in this sentence; line 130. – Instead of “generating”, “generation” would be more acceptable; line 160. – “…lead to reduction of NO…”, line 329. – “…preventing myocardial infarction injury…” etc.
  2. Line 130, line 301, line 307, line 320, line 385, line 450 - those sentences need to be rephrased, because they are either grammatically or logically incorrect

Author Response

Dear reviewer,

We would like to thank you for reading our manuscript again. Our manuscript was read by a native English-speaker and we corrected errors suggested by you.

Answers to major comments:

  1. Line 28 – Acut myocardial infarction is the most severe complication of coronary artery disease. In this review there are two different prevalence for AMI and CAD, which is incorrect.

We specified and corrected the sentence meaning. The sentence is rewritten as follows “Of total cardiovascular deaths, coronary artery disease is the leading cause at 60%, of that acute myocardial infarction (MI) comprises 25%”.

  1. The explanation under the figures are missing (except of figure 2., but the text under the picture is crossed out)

We are really sorry that the latest version that was resubmitted appears to be an English pre-edited version as well as our correction and added parts were omitted. Therefore, the explanation under the pictures were either missing or crossed out.  We double check all your comments and points and edit the latest version in the system.

  1. The NO can have either beneficial and detrimental biological effects. The authors state that all isoforms of NOS are activated due to ischemia, but later there is a statement which declare that NO level is decreased in early stage of reperfusion. During the reperfusion period the oxidative stress has a pivotal role in the IR injury, and higher NO may have crucial role this process with the formation of peroxynitrate. Taking everything into account, the role of NO in myocardial infarction and IR injury is still not clear in this review.

We agree with reviewer that nitric oxide can have ambivalent effects and multiple studies have shown discrepancies. From the sentence we used it was not clear what we ment. Since we found only one paper which described decrease of NO level during early phase of reperfusion only (not after couple hours), we deleted this sentence to avoid misinpretation.

Minor comments:

  1. There is still a numerous linguistic error. E.g.: line 54. – “…caused by the formation…”; line 58 – the word “mechanism” should be used in plural; line 59 – the word “cardiomyocyte” should be used in plural; line 69 – “The intracellular concentration of sodium ions is significantly increased and it/this phenomenon/this process activates the Na+-H+ ion exchanger – without a noun the sentence is not correct.; line 73. - instead of “with” the writers should use “resulting”/”as a consequence”/”as a result” because the Na overload is a consequence of poor 3Na+-2K+ ATPase function; line 75 - the expression “main source” is incorrect again, the authors should use e.g. “main signal transduction pathway” etc.; line 78. – “elevates”; line 107. – the word “pathway” is superfluous in the end of the sentence; line 112. - the word “establishment” is incorrect in this context, “formation”, “generation” would be preferable; line 118 – the word “tissue” is crossed out in the text notwithstanding that it is necessary in this sentence; line 130. – Instead of “generating”, “generation” would be more acceptable; line 160. – “…lead to reduction of NO…”, line 329. – “…preventing myocardial infarction injury…” etc.

The sentences are rewritten as follows:

Lines 49-51 (54): When the plaque endures a rupture, the disruption of blood circulation by reduced perfusion and/or occlusion of the coronary artery causes by the formation of a thrombus.

Lines 52-53 (58): When the heart survives the ischemia, either an adaptive mechanisms ...

Lines 63-66 (69): The intracellular concentration of sodium ions is significantly increased and this process activates the Na+-H+ ion exchanger [14]. The lack of ATP during ischemia discontinues the function of 3Na+-2K+ ATPase withas a consequence of an intracellular Na+ overload.

Lines 70-71 (73, 75): ... has been identified as a main signal transduction pathwayfactor in cardiomyocyte apoptosis...

Lines 72-73 (78): Moreover, the upregulation of phosho-p38MAPK increases cardiomyocyte apoptosis,...

Line 102 (107): the word pathway was removed

Line 123 (130): Reperfusion is a multifactorial complex that involves i) generationng reactive oxygen species...

Lines 310-311 (329): A therapeutically relevant strategy for preventing myocardial infarction injury is to restore blood supply to the ischemic area and minimize related damage.

  1. Line 130, line 301, line 307, line 320, line 385, line 450 - those sentences need to be rephrased, because they are either grammatically or logically incorrect

The sentences were corrected are re-writte as follows:

Lines 294-307 (301-307): Non-flavonoids are found in different kinds of citrus fruit and berries, coffee, olive and sesame oil, cereals, and red wine. The most important class of non-flavonoid polyphenols is phenolic acid. Both in vivo and in vitro studies have shown their antioxidant and anti-inflammatory characteristics due to their ability to reduce the production of inflammatory markers, such as IL-1β, IL-8, MCP-1, COX-2, and iNOS [99]. Phenolic acid can protect cells against various injuries depending on their chemical structures [100]. The main representatives of phenolic acids are derivatives of benzoic or cinnamic acids [101]. These acids include gallic acid, vanilic acid, caffeic acid, and ferrulic acid (Table 1). Natural stilbenes are characterized by the presence of a 1,2-diphenylethylene nucleus [102]. They have low bioavailability [103] depending on the route of their administration, absorption, and metabolism. Their metabolism is determined by their chemical structure, molecular size, degree of polymerization, etc. [104, 105]. In addition to their antibacterial effects, stilbenes also exert anti-hypertensive and anti-tumor effects, as well as inhibit platelet aggregation [106] and prevent the cell damage induced by oxidative stress, hypoxia, and ischemia [107-110]. Resveratrol is one of the primary representatives of stilbenes.

Lines 310-311 (320) : A therapeutically relevant strategy for preventing myocardial infarction injury is to restore blood supply to the ischemic area and minimize related damage.

Lines 373-375 (385): An experimental study identified the different molecules and pathways involved in the protective effects against ischemia/reperfusion injury.

Lines 438-442 (450): In addition, EGCG administration can ameliorate apoptotic markers, such as Bax and caspase 3 and 9, accompanied by the inhibition of DNA fragmentation and the downregulation of p53. EGCG treatment was able to attenuate MI by reducing the infarct size in Langendorff perfused rat hearts by acting on the adenosine receptor (ADR) and opioid receptor (OPR).